# Possible Perception Bias in the Thermal Evaluation of Evaporation Cooling with a Misting Fan †

Craig Farnham [1],* and Jihui Yuan [2]

1 Department of Human Life Science, Osaka City University, Sumiyoshi-ku Sugimoto 3-3-138, Osaka 558-8585, Japan
2 Department of Architecture and Civil Engineering, Toyohashi University of Technology, 1-1 Hibarigaoka, Tempaku-cho, Toyohashi 441-8580, Japan; yuan.jihui.pz@tut.jp
* Correspondence: farnham@life.osaka-cu.ac.jp
† This paper is an extended version of our paper published in Proceedings of the 10th Windsor Conference, Windsor, UK, 12–15 April 2018; pp. 96–106.

**Abstract:** Mist evaporation cooling (MEC) is increasingly used as a low-energy means to improve thermal comfort in hot environments. However, the thermal sensation votes (TSV) often overshoot values of Predicted Mean Vote (PMV) models. Evaluations of MEC may be affected by an expectation that mist feels cool or the "good subject" effect. Here, subjects are exposed to a misting fan and an identical fan without mist and asked which fan feels cooler. Unknown to the subjects, the misting fan has almost no cooling effect (about 0.4 K reduction in air temperature) and a hidden heater increased the temperature of the misting fan air flow, making it up to 1.6 K warmer than the fan without mist. Supplemental experiments told the subjects about the heater. Surveys of over 300 subjects when varying this misted air temperature showed a bias above random chance that people vote that a misting fan airflow was cooler, even when it was the same temperature or slightly warmer than the non-misting fan. It is possible that the expectation of cooling or good subject effect influences evaluations of mist. This effect should be considered in thermal comfort evaluations of mist cooling and in the deployment of MEC systems.

**Keywords:** mist; evaporation; cooling; thermal comfort; expectation; PMV



## 1. Introduction

Our world is getting hotter and the need for energy-efficient cooling strategies is increasing. Hot conditions and heat waves can adversely affect human health [1], with increasing mortality as the heat index increases [2]. Even if heat stress does not directly provoke illness, performance and productivity generally decrease at effective temperature over 30 °C [3]. As the risk of heat stroke increases exponentially at higher temperatures [4], even a small reduction in temperature could be highly beneficial in hot environments.

In the face of the COVID-19 pandemic, there has been a shift to outdoor work and activity, and a greatly increased use of Personal Protective Equipment (PPE) by workers [5] and masks in the general population. These are generally not designed for prolonged use in hot outdoor environments. [6] There may be even further increase in demand for means to reduce thermal stress in outdoor environments.

There will be increasing demand for cooling strategies that can be deployed outdoors to help prevent heat stroke. The Centers for Disease Control and Prevention (CDC) recommends heat illness prevention methods, such as the use of cooling or misting fans at cooling stations [7], provided the air flow does not blow respiratory droplets from one person to another.

Water evaporation has been used for millennia in arid climates [8] as a means of cooling. It is well-suited to reducing air temperature in hot environments at low or no energy cost. The major drawback is that as air temperature is reduced by the evaporating

water, its humidity increases. The limit is saturated air at the wet bulb temperature. Thus, the potential for cooling is better in hot, dry environments. Mist evaporation cooling (MEC) typically uses high-pressure pumps to produce a water mist with droplets of average diameter well under 100 microns. This increases the surface area between water and air by several orders of magnitude more than water in a pool, such that the water quickly evaporates even in humid climates. Droplet temperature quickly reaches the wet bulb temperature, on the order of tens of milliseconds, and evaporates completely on the order of seconds. When using high-pressure hydraulic spray nozzles, the net cooling effect (the energy of evaporation of the sprayed mist) can be well over 100 times greater than the energy consumed by the high-pressure water pump, making MEC an attractive energy-saving method to provide cooling where the running costs of standard vapor-compression air conditioning would be prohibitive.

As evaporation continues and air vapor pressure increases, the droplet evaporation rate slows. If the flow of mist and air reaches a surface before the droplets evaporate, there is a risk of wetting. This could be a slipping hazard, cause property damage or yield discomfort or complaints from persons. Thus, in practice MEC for thermal comfort is often used in open or semi-enclosed spaces, where the natural flow of outdoor air helps dilute the effect and prevent saturation. In open air applications, it is desirable to arrange nozzles such that surfaces in the MEC area will not be significantly wetted. This is possible even in sub-tropical climates such as much of Japan [9], which has a similar climate to the southeastern United States, much of eastern China, and much of the southeast coast of South America [10]. Mist can even be used effectively in the tropical climate of Singapore [11]. Mist cooling can cause undesirable wetting if used indoors. An example of a clean room of 40 m$^2$ floor area with about 200 m$^3$ volume in Japan, ventilated at approx. 2 air exchanges per hour, a mist fan at 3.5 m height spraying 5~20 L/h of mist yielded some floor wetting [12] which could be a slipping hazard.

### 1.1. Evaluations of Mist Evaporation Cooling

MEC thermal comfort experiments have shown great improvements in thermal sensation votes (TSV). Ulpiani et al. [13] surveyed 211 people during the summer at two locations in Italy with overhead misting installations. At one location, where the average air temperature drop under mist was 2.8 °C, 81% of respondents had a TSV of +2 or +3 without mist, while 63% then voted −1 or −2 with mist. The number of persons wishing wetter conditions "paradoxically" increased among those experiencing mist. In the urban location of Rome where the average temperature drop was 1.6 °C, over 50% had TSV of +2 or +3 without mist, and over 90% in the comfortable range of −1 to +1 within mist.

Oh et al. [14] surveyed 65 subjects during summer in Tokyo under an overhead misting system with added blown air. With combined mist and blown air yielding a 3.6 °C air temperature drop, average TSV decreased by 1.7 steps on average to a neutral 0.0. This drop was 1.4 steps without blown air and a 2.9 °C temperature drop, and 0.9 steps with blown air but the amount of mist sprayed was reduced by 20% yielding a 1.9 °C temperature drop.

Wong et al. [11] found that a misting fan system in the tropical climate of Singapore reduced TSV by 1 step with an average temperature drop of 1.5 °C, while an overhead misting line (without fans) for a rest space did not significantly change the air temperature or TSV, likely due to the misted air not reaching the space. Under the mist fan, a regression of TSV against Effective Temperature (ET *) showed thermal neutrality could be obtained at 0.8 °C higher outdoor ET * with the mist fan system.

Kohno et al. [15] combined MEC with sunshades of various transparency for outdoor comfort. Test subjects sat under each type of sunshade for 5 min and completed comfort surveys with and without MEC. Under MEC in direct sunlight without shade, there was little comfort improvement. Responses of "uncomfortable" were 81% without MEC and 80% with it. When shaded, "comfortable" votes were much improved, exceeding 90% in some cases. More transparent shades tended to have lower comfort responses without MEC, by approximately the same as the fraction of blocked sunlight. In the least-shaded

case (10% of sunlight blocked, 7% comfort without mist), addition of MEC increased votes to 63% comfortable.

Miyasaka et al. [16] performed physiological measurements and comfort surveys for 7 subjects for the "Zero Energy Cool Tree" system. The "tree" is a wooden platform and sunshade with a central column. MEC spray nozzles are mounted in the shade to create a cool space. Water-cooled wooden benches (at surface temperature of 20 °C or 24 °C) are under the shade. On a humid summer day (32 °C air temperature, 55% humidity), the air temperature at the benches was about 0.5 °C cooler than the surrounding air. The average TSV for cooled benches without mist was reduced from about +1.8 to +0.5 without mist, and to +0.2 with mist.

Nakai et al. [17] used mist with Sauter mean diameter (SMD) of 20 microns as a supplement to spot air-conditioning (AC) in a factory. The factory was ventilated at 2 air changes per hour. Spot AC with and without MEC at 2 air velocity settings was evaluated. At 1.5 m height, air temperature with MEC was 1~3 °C cooler than AC alone. The relative humidity increased 10~25%. Standard Effective Temperature (SET *) was 1~2 °C lower depending on the air velocity setting. About 20% of workers experienced a 1-step improvement in both thermal comfort (on a 7-step scale) and overall comfort (on a 5-step scale) when MEC was activated. However, there was a 1-step higher feeling of humidity (on a 5-step scale).

In the humid sub-tropical climate of Japan, as designated by the Köppen climate classification system [18], MEC was deployed at the 2005 World Expo in Aichi to improve thermal comfort [19]. Overhead mist sprayers in a shaded waiting area reduced the air temperature by up to 3 °C on hot summer days. Surveys of 1424 people using the upper half of the standard PMV scale from +1 (slightly warm) to +3 (hot), with the last option being "not hot", about 75% of responses were for +1 or "not hot" when mist cooling was strong, while over 60% of Reponses were for +2 or +3 when cooling was not in effect. In a different project, downdraft MEC was used on a train platform in Japan, yielding reduced about 1–3 °C reductions in air temperature. Surveys showed over 80% claimed improved thermal comfort [20].

Combination of MEC with fans can help to direct the cooled air in the desired direction. In the tropical rainforest climate of Singapore, the use of fans fitted with high-pressure atomization nozzles to produce fine droplets improved thermal comfort votes more than the effect from fans alone, while overhead mist spray without a fan did not [11].

Furumoto et al. [21] used overhead MEC at a train station concourse and gate. They conducted extensive surveys of comfort, as well as the riders' aesthetical opinion. MEC was compared to standard air conditioning (AC) and natural ventilation over 2 months. The MEC area had average drops in SET * of 0.5 ~ 1.5 °C. In overall satisfaction, MEC slightly outranked AC and natural ventilation. In the evaluation of aesthetics, MEC was highest at 63% positive vs. 41% for AC. The survey also asked subjects if they had noticed MEC was operating. The 59% of those surveyed that did not notice the mist was operating were twice as likely to vote at +3(very hot) on the TSV scale and twice as likely to vote "uncomfortable" on a 7-level TSV scale. Those who noticed the mist tended to wish the MEC effect was even stronger. These results indicate that perception of MEC may affect thermal comfort votes.

Our previous research on MEC with fans also yielded results that indicated the possible over-evaluation of MEC by comparing the TSV of exposure to a MEC fan to the same fan while not spraying mist. A survey of 158 people showed the average TSV improvement from the fan alone on a hot summer day in a humid climate in a shaded space (33–35 °C, 44–50% RH) was about a 3-step reduction, while the MEC fan was about 4.6 steps cooler [22]. The air velocity was unchanged, but the air temperature ranged about 2–3 °C cooler in the MEC air stream.

TSV in that experiment with human subjects in outdoor MEC were far better (cooler) than the ASHRAE 55 [23] thermal comfort Predicted Mean Vote (PMV) models predict (as shown in Table 1, taken from [22]). This can be expected, as standard thermal comfort

models and the 7-step TSV scale were not designed for outdoor, high-temperature use, nor for a sudden thermal change [24].

**Table 1.** Results of previous comfort experiment research comparing a fan to a mist evaporation cooling (MEC) fan.

| Condition | Air Temp. | Rel. Humidity | Air Velocity | PMV | Avg TSV |
|---|---|---|---|---|---|
| Base condition, no fan | 35 °C | 50% | 0.1 m/s | +3.23 | +2.98 |
| Fan only | 35 °C | 50% | 2.7 m/s | +1.37 | +0.05 |
| MEC fan (2 K drop) | 33 °C | 58% | 2.7 m/s | +0.86 | −1.65 |
| MEC fan (3 K drop) | 32 °C | 63% | 2.7 m/s | +0.61 | |

Many of these results show improvements in TSV beyond what would be expected by the standard model of Predicted Mean Vote (PMV) model. Plots of Effective Temperature (ET) on a psychrometric chart tend to decrease in value along lines of constant enthalpy following the progression of air temperature in the adiabatic process of MEC [24]. However, these decreases in ET are relatively small.

A likely major cause of this over-evaluation is the thermal transient effect, as extensively evaluated by de Dear et al. [25]. A sudden change in temperature tends to lead to an over-evaluation of thermal comfort, which then reverts to a more moderate evaluation over the span of a few minutes to an hour. This over-evaluation is doubly strong when moving from warm to cool than the opposite. Given that mist cooling experiments are often performed outdoors in short terms on hot days, this thermal transient over-evaluation effect should tend to the same strong reaction.

MEC has a unique aspect compared to typical conditioned air and convection cooling with fans. Mist is visible. In our experience while building and testing mist systems in the off season, visiting staff and students would claim that mist felt cool even on mild days, or when the mist spray flow was quite weak and non-wetting (when meant for humidification, not cooling), or when the air temperature drop was less than 0.5 °C. It seemed that people expected a spray of water mist to be cool, and this may affect their evaluations.

A single factor sensitivity analysis [26] of the PMV for each of the 6 inputs (air temperature, radiant temperature, air velocity, vapor pressure, clothing insulation and metabolic rate) showed that the PMV should decrease by about 1 step for a 10 °C decrease in temperature or a 10 °C decrease in radiant temperature, while PMV should decrease by about 0.1 step for a 700 Pa decrease in vapor pressure. However, the previously mentioned MEC comfort experiments tend to show TSV improvements of 1 or 2 steps or even more, with temperature drops of well less than 5 °C.

*1.2. Human Perception and Thermal Comfort Evaluation*

There is much research on perception of thermal comfort in step changes in air temperature, but these are step changes on the order of 5 °C or more [25,27,28], which of course yield significant changes in sensation. To our knowledge, the only research focusing on a comfort step change "control" in which perception is tested without an actual step change, are experiments which use deception. Rohles' [29] experiments on thermal comfort showed that two rooms of identical thermal environment were evaluated differently based on visual cues. When one room was visibly outfitted as a walk-in refrigerator (though not in use as such) to present a "cool" image, subjects rated it as cooler. When the room was redecorated with wood paneling, carpets and such, presenting a "warm" image, the room was evaluated as warmer than the control case. Stramler et al. [30] showed that experiment subjects in a room cooled by 2.8 °C had the same improvement in comfort votes as those in a room without cooling where a thermometer was falsely indicating 2.8 °C of cooling to the subjects.

Human skin can detect quite small changes in temperature if they are relatively fast. Stevens and Choo [31] found the minimum rapid step change detectable by the glabrous skin of the palm near the thumb is 0.20 °C for an increase and 0.11 °C for a decrease.

However, those experiments were by thermal conduction from a Peltier device in contact with the skin. The forced convection of mist-cooled air may be difficult to directly compare to that.

There is a large body of research on the diagnosis of fever by touch (by clinicians or by untrained people). The goal is to determine if touch can yield an accurate yes/no diagnosis for the relatively small temperature deviation between normal body temperature and fever, while there are possible confounding biases (other symptoms, concern for patient, etc.) These statistical analyses tend to focus on sensitivity (correctly finding positives) vs. specificity (correctly judging negatives) and then the power as a clinical diagnostic tool, given the likelihood of fever. Temperature itself is not used as an independent variable, rather the presence or lack of fever is determined by a chosen cutoff temperature. A study by Singh et al. [32] on determining presence of fever (over 37.5 °C) by touch showed a family member could accurately detect a patient's fever in 62% of cases, while clinician was correct in 71% of cases. They determined that judgment by touch had high sensitivity but low specificity, and thus judgment by touch is incapable of making large and clinically meaningful results. Whybrew et al. [33] determined that "as a screening procedure, touch will seriously overestimate the incidence of fever, but with touch, fever will rarely be missed". Generally, judging fever by touch has high sensitivity but low specificity. If fever is present it is usually detected, but if there is no fever, there are many false positives, thus it is not seen as an appropriate diagnostic. A test for bias to over-evaluate MEC could check for specificity (correctly judging the negative) when the MEC is not actually significantly cooler than ambient.

The "hue-heat" hypothesis considers that the visual stimulus of color might influence the perception of temperature [34]. In terms of thermal comfort, Fanger et al. [35] found that extreme red light yielded a slight decrease in preferred environment temperature compared to extreme blue light, but not to any practical significance. Mosley and Arntz [36] found that using a red color cue produced much stronger evaluations of hot (as much as 5.5 points higher on an 11 point scale) and painful sensations than a blue colored cue when touching a cold rod with one hand for 500 ms. In research testing the sense of touch of a hot or cold cup with the localized visual cue of a blue or red cup in a virtual environment [37] it was shown that reactions trended toward feeling cooler for blue and warmer for red. The white/bluish color of a misting fog might also be perceived as a cool color.

Those participating in a mist cooling experiment might be influenced by the "good subject" effect [38], a desire to give the "correct" answer which validates the experiment hypothesis. Subjects may seek overt or implicit cues to determine what the "correct" answer should be and give that answer. The motives can include a desire to help the researcher, hope to aid the progress of science, seek approval of the experimenter, or other motives. The experiment goals might also be a subject of rumor among the population from which subjects are taken. In the case of mist cooling on hot summer days, it is possible that test subjects would expect that the "correct" answer should be to claim that mist feels cool. If subjects are recruited from the student population at the campus site, rumor and foreknowledge of the experiment may influence responses.

### 1.3. Research Objective

The pattern of strong improvements in TSV for relatively small temperature drops under MEC may in part be due to over-evaluation as a result of a perception bias. Such over-evaluation may be based on an expectation that mist is cool, the "good subject" effect, or other causes. To our knowledge, there are no blinded outdoor MEC evaluation experiments in the literature. The purpose of this study is to use deception, as has been done previously in thermal evaluations, as a form of experimental blinding to determine if the responses of subjects in a MEC thermal comfort evaluation may tend toward evaluating mists as cooler than the physical reality.

Subjects will be presented with MEC-cooled air and air at the same temperature without MEC and asked to judge which is cooler.

Of note, if there is a bias to over-evaluate MEC, this study would work to lessen our lab's positive results in reports of mist cooling in hot summer outdoor conditions from hundreds of test subjects over the years. However, it may lead to a more accurate re-evaluation of past results, better experiment design from ourselves and others in this field, and better design of MEC systems.

## 2. Experiment Design

The core concept of this experiment is, "All other thermal factors being equal, do people tend to choose a visible mist as cooler than dry air?" The key to this experiment design is to present a comparison of visibly misted air to non-misted air, but ensure both have same thermal effect on human subjects. This also ensures the cooling thermal transient effect is not a factor in the comparison, as the transient is the same.

Comparing a mist spray to still air of the same temperature would be difficult, as the mist spray momentum entrains surrounding air to yield a stream of moving air. The misted air thus includes some forced convection cooling, which differs from natural convection in still air. Thus, a stream of moving air, compared to a similar stream of moving air with visible mist was chosen as a means to compare mist against non-misted air. If both air streams have the same velocity and temperature, the cooling effect should be the same, and the human evaluation should be the same. This was accomplished by using two misting fans of the same make and model, activating the mist function on one to produce a visibly misted air stream (VMAS) and leaving the misting function off for the other to produce only the fan air stream (FAS). The VMAS can be heated by some means to negate the evaporative cooling effect, and bring its temperature up to the FAS temperature, or even higher. This means must not be evident to the test subjects, so a hidden device behind the fan to pre-heat the fan inlet air was chosen. Care must be taken to ensure there is no wetting of the test subjects' skin. This would introduce another source of cooling, contact evaporation and/or conduction with a liquid film.

Subjects are asked to judge the VMAS and FAS with their bare hands, which are known to have good temperature sensitivity as per Stevens and Choo. This localized evaluation helps eliminate clothing as a factor, and allow simultaneous evaluation of the 2 air streams, eliminating any time lag during a step change as a factor.

If possible, subjects from the general public can be recruited to reduce the possible influence of "campus rumor" as part of the "good subject" effect.

### 2.1. Experiment Goals

The experiment presents a choice of "Which feels cooler, mist or non-mist, or are they the same?" If subjects can accurately judge the temperature difference, a VMAS and FAS at the same temperature should tend to yield a choice of "Same".

Here, we handle this as similar to the research on judgment of fever by touch, as a test of "specificity" in the diagnostic sense. Do the subjects correctly evaluate the VMAS and FAS of the same temperature as the same, or not? That is used to create the first null hypothesis.

Among the responses that are not for "Same", if there is no bias to choose mist, there should be a random (50%/50%) split among the remaining votes for VMAS or FAS as cooler. If a significant majority of subjects choose VMAS as cooler, then we could take this as evidence of a bias. This is used to create the null second hypothesis.

Lastly, we consider that the experiment setup can also yield cases where the VMAS actually is warmer. We test the choice of either the VMAS or "Same" instead of FAS. If VMAS is actually warmer, then choices of VMAS or "Same" as cooler are incorrect. If more common than random chance, it could be taken as evidence of a bias to perceive mist as cooler than the physical reality. This is used to create the third null hypothesis.

Three null hypotheses $H_0$ are examined:

1. $H_{0,s}$: When comparing a VMAS to a FAS of similar temperature and velocity, given 3 choices (VMAS, FAS, or "Same") more than 50% of subjects will (correctly) evaluate them as the same.
2. $H_{0,m/f}$: When comparing a VMAS to a FAS of similar temperature and velocity, among those who do not choose "Same", no more than 50% will choose the VMAS as cooler.
3. $H_{0,ms}$: When comparing a VMAS to a FAS of similar temperature and velocity, given 3 choices (VMAS, FAS, or "Same") no more than 67% will choose the VMAS as cooler or "Same".

All results, as proportions of A/B choices among the subject population, are evaluated for statistical significance against a binomial distribution. This initial study does not seek to determine the cause of a bias, if any, to favor mist as cool. It may be an issue of perception, psychology, physiology, "good subject" bias, a combination of these, or other factors. The goal is only to see if a bias might exist. Investigating the causes will require a much more detailed experiment program, including the measurement of subjects' skin and body core temperatures, evaluation of thermal history, interviews on personal opinions of mist cooling, and more. Here, we chose to survey hundreds of test subjects in interactions taking about 5 min, rather than a handful of subjects in detailed observation over the span of hours, as repeated use of the same subjects increases the chance for the subject to discover the deception.

Furthermore, this study does not seek to define a numerical value for the possible over-evaluation of mist cooling at some sort of set temperature limit. As an example, if this study finds that in these particular experiment conditions, that this VMAS will still be chosen as cooler by the majority of subjects until it is more than 1 °C warmer than the FAS, that would have little general application to all mist cooling experiments. However, it may serve as a hint as to the range of temperatures that should be tested in further studies.

### 2.2. Air Temperature Reductions in Mist

To design a system that produces a VMAS and a FAS of equal blown air temperature, the effect of heat exchange between mist and air must be evaluated. Mist cooling is a constant-enthalpy process. The latent heat of evaporation is exchanged with the air's sensible heat, reducing air temperature. The air condition follows a constant-enthalpy process, with deviations due to any sensible heat change of the liquid water. This deviation is less than 5% of the net heat of evaporation even if the supply water temperature is 30 °C warmer than the air temperature. A 5% reduction in a small cooling effect (on the order of 1 °C) would yield an immeasurable change (with the sensors used here), thus it is ignored.

The air temperature change is a function of the mass of evaporated water, $m_{ev}$, and the amount of air with which it interacts, $m_{air}$. Increased mist flow will yield increasingly lower temperature and higher humidity in this adiabatic process. Given enough mist, the limit of saturated air at the wet bulb temperate and 100% relative humidity may be reached. The net effect is a balance between the latent heat of evaporation of the mist and the sensible heat change of the air,

$$m_{ev}L = m_{air}\, C_P\, \Delta T_{av} \tag{1}$$

where $L$ is the latent heat of evaporation of water, $C_P$ is the specific heat of air and $\Delta T_{av}$ is the average temperature drop of the air.

In practice and in this experiment, fine mists will typically be deployed to evaporate completely, thus $m_{ev}$ is simply the spray flow rate. In a duct or closed space, the mass of air is easily known. However, in the case of an open-air fan, where blown air entrains surrounding air, the amount of air with which the mist interacts cannot readily be evaluated, making accurate predictions of temperature drops difficult. In the case of a misting fan, one may assume the "best case" of average air temperature drop is if only the amount of blown air, $m_{ba}$ from the fan interacts with the mist.

$$\Delta T = (m_{ev}\, L)/(m_{ba}\, C_P) \tag{2}$$

The actual average temperature drop will decrease in proportion to the amount of surrounding air entrained into the blown air.

### 2.3. Experiment Apparatus

As the goal of the experiment is to test a VMAS against a FAS of about the same temperature, a misting fan with a small cooling effect was chosen. Two home-consumer oriented fans of the same model with built-in ultrasonic mist generators with average droplet diameter of under 10 μm were used. In the environment conditions in these experiments, mist droplets under 10 μm tend to evaporate on the order of well under one second as can be calculated by the iterative methods presented by Chaker et al. [39] and Holterman [40], thus yield little or no surface wetting.

The mist spray rate was measured by the change in weight of the water supply tanks after misting continuously for 3 h. The average cooling effect of evaporation is determined from the spray rate and the latent heat of evaporation of water, assuming complete evaporation (as confirmed by observation) as the left side of Equation (1). The fans' blown air volume was measured by attaching a circular duct to the fan outlet an applying the flow measurement technique as per ASHRAE-111 [41]. This is used as the blown air flow in Equation (2) to find the expected average temperature drop due to mist cooling $\Delta T$. The expected average humidity increase is the ratio of mist sprayed to blown air, at about 0.012 g/kg dry air, which equates to less than a 0.5% increase in relative humidity at 30 °C.

The fan electricity consumption was measured with a real-time Watt meter at the electrical outlet while the fan was on and while the fan and mist spray were both on. The fan outlet grills include directional vanes and were set to rotate during all experiments (at 1 cycle per 25 s), to help diffuse the mist over a wider cross-section, further reducing the possible localization of temperature drop. These fan characterizations are listed in Table 2.

**Table 2.** Characteristics of the misting fans.

| Specification | Fan A | Fan B |
|---|---|---|
| Spray rate | 165 mL/h | 171 mL/h |
| Blown air, medium setting | 0.20 m$^3$/s | 0.21 m$^3$/s |
| Air velocity at fan centerline, 50 cm dist. | 2.6 m/s | |
| Outlet grill directional vane rotation | 1 cycle/25 s | |
| Power consumption, fan only | 46 W | |
| Power consumption, fan and mist | 78 W | |
| Evaporative cooling effect | 112 W | |
| Expected average temperature drop | −0.5 K (0.4 K with mister waste heat) | |
| Expected average humidity increase | +0.012 g/kg (DA) | |

At the fan speed setting used here, the expected average temperature drop of the misted air stream is 0.5 K. Assuming the added electric consumption of the mist spray function (32 W) becomes waste heat added to the blown air, it offsets some of the cooling effect. If included in the heat balance calculation, the average temperature drop reduces to about 0.4 K. The two fans' spray rate and blown air rate differed by under 5%. This should have no measurable effect on the average temperature drop.

The misted air flow was heated by a 10 m length of 10 mm diameter copper pipe wound into a loose coil of about 25 cm diameter with circulating hot water mounted behind one of the fans as shown in Figure 1. By adjusting the temperature and flow rate of the water, the blown air temperature could be increased up to about 2 K above the ambient. For ease of operation, in the third and following experiment trials the hot water coil was replaced by two 300-Watt electric radiant heaters which yielded a similar temperature range. In all trials, the systems were turned on and off at operator discretion at 15 or 30-min intervals to yield a range of temperatures. These devices and sensor outputs were

all shielded from the view of the test subjects by a black shield made of "pladan", sheets similar in structure to corrugated cardboard but made of plastic.

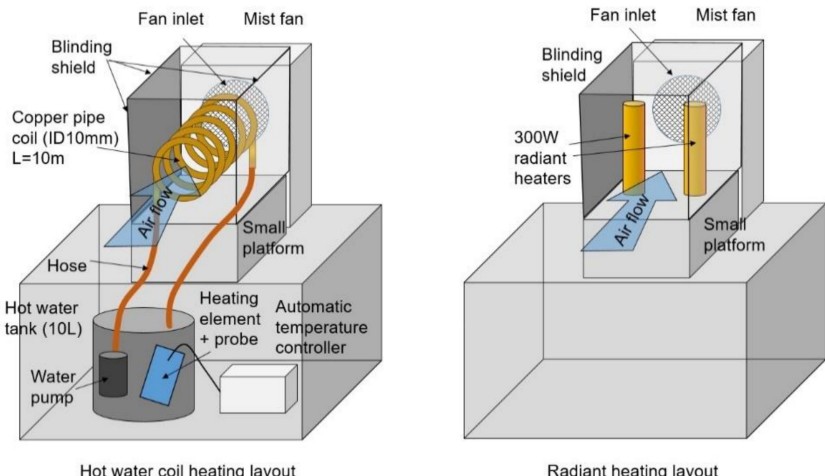

**Figure 1.** Layouts of heating systems placed behind the visibly misted air stream (VMAS).

## 3. Experiments

The experiments performed include a characterization of the spatial distribution of temperature drop in the VMAS and the surveys of experimental subjects while measuring temperature along the VMAS and FAS centerlines only. These thermal evaluation surveys took place over 3 trial periods with blinding (not revealing the VMAS has been heated), then 2 supplemental trial periods with experimental blinding eliminated.

### 3.1. Temperature Measurements and Apparatus Characterization

The temperature of the air streams was measured with thermistor temperature loggers with a rated measurement error of +/−0.5 °C in the first and second evaluation experiment trials. Before the experiments, 6 of these loggers were calibrated over a span of 30 min in the same environment as the experiment trial. The two that maintained the most uniform reading (within 0.2 °C of each other at all times) were used to measure the VMAS and FAS temperatures. The sensors were placed 50 cm from the fans, along the centerline of the fan axis. In all experiments, the mist produced by the fans was seen to completely evaporate within about 25 cm of the fan, thus there was little chance the sensors were wetted.

Starting with the third trial, T-type thermocouples connected to a data logger were used to measure the air stream temperature. They were calibrated similarly, with the two thermocouples used always reading within 0.2 °C of each other, which is taken as the calibrated measurement error.

As the basis of evaluation is the temperature difference between the two air streams, the maximum error is assumed as the "worst case" as the sum of both errors in opposing direction, so +/−0.4 °C according to calibration, and +/−1.0 °C according to the manufacturer rated accuracy.

As the mist emitter is located at the center of the fan, the local air temperature along the fan axis should be lower than the average expected air temperature calculated by Equation 2. An array of 5 T-type thermocouples (as used in Trials 3, 4 and 5) and 3 thermistor temperature loggers (as used in Trials 1 and 2) was arranged in a 10 cm × 10 cm arrangement similar to the shape of a human hand and placed at 50 cm distance along the fan centerline to measure the local temperature while spraying mist, using the heaters, or both together. The fan was run with no added effects to establish the base average environment temperature, then with the heater activated at 300 W for 5 min and returned to no added effect for 5 min to yield a clear differentiation. This no effect/added effect/no effect 5 min cycling was repeated with the heater at 600 W, twice with no heater and only mist (Mist, Case 1 and Mist, Case 2), mist and 300 W heater, and mist and 600 W heater.

The expectation is that the mist and heater cases should have zero or change or even an increase above the ambient temperature case.

An example of temperature data is shown in Figure 2. The temperature change resulting from mist or heating or both is found by interpolating the average air temperature for the 3 min before and after the change, and comparing to the average temperature during the change, where the average values are represented by "X" marks. Notice that the oscillation in thermocouple temperature at each sensor matches the period of the outlet vane rotation (1 cycle per 25 s).

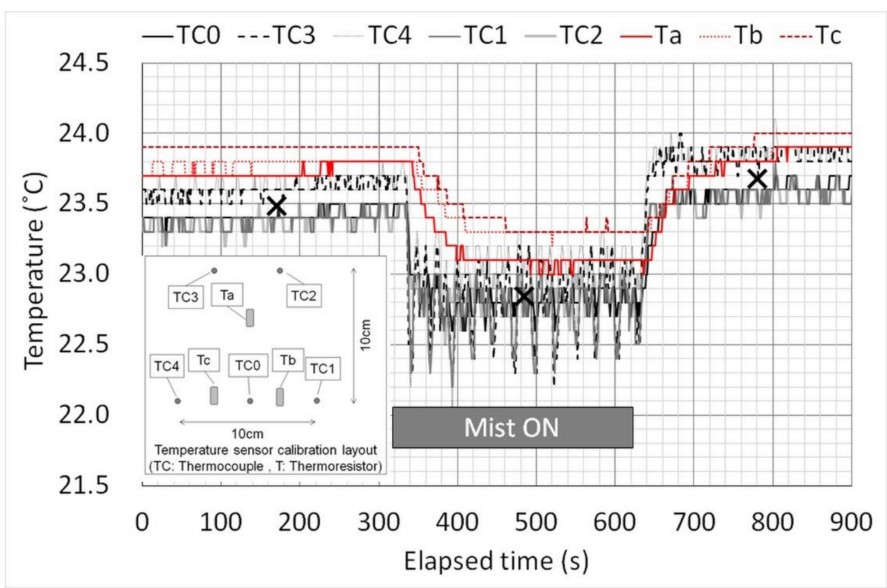

**Figure 2.** Example of misting fan and heater characterization temperature measurements. Here, misting without a heater. Sensor positions shown in insert at lower left.

The average temperature changes within this 10 cm × 10 cm area are shown in Table 3, with thermocouples listed as "TC" and thermistors as "T". The thermocouples recorded a temperature slightly lower than the thermistors. Among each type, the recorded temperatures never exceed more than +/−0.2 K difference from the average of all sensors of that type, with the exception of the sudden change when misting was activated. The 80% response time of the thermocouples was under 3 s for the sudden change of mist activation. The 80% response time of the thermistors was about 60 s or less after the sudden change of mist activation. Over the long term of the hours-long experiment trials, this time delay is likely not a factor. As the temperature difference between Fan A and Fan B is based on the difference between two calibrated sensors of the same type, the maximum error in temperature difference should be +/−0.4 K.

**Table 3.** Measurements of misting fan and heater temperature changes (in K) of thermocouples (TC) and thermistors (T).

| Scheme 0 | TC0 | TC1 | TC2 | TC3 | TC4 | Avg | ΔT(TC) | Ta | Tb | Tc | Avg | ΔT(T) |
|---|---|---|---|---|---|---|---|---|---|---|---|---|
| Avg.Environment temperature | 23.9 | 23.9 | 23.9 | 24.1 | 24.2 | 24.0 | - | 24.1 | 24.1 | 24.3 | 24.2 | - |
| 3300 W heater | 24.8 | 24.6 | 24.7 | 24.9 | 24.8 | 24.8 | +0.99 | 25.1 | 25.2 | 25.3 | 25.2 | +1.03 |
| 600 W heater | 26.4 | 26.6 | 26.7 | 26.6 | 26.9 | 26.7 | +2.54 | 26.7 | 26.5 | 26.9 | 26.7 | +2.52 |
| Mist, Case 1 | 22.8 | 22.8 | 22.9 | 22.8 | 23.0 | 22.8 | −0.74 | 23.1 | 23.3 | 23.3 | 23.2 | −0.67 |
| Mist, Case 2 | 24.1 | 23.9 | 24.2 | 24.1 | 24.1 | 24.1 | −0.70 | 24.3 | 24.6 | 24.7 | 24.6 | −0.64 |
| Mist and 300 W | 25.8 | 25.9 | 26.1 | 25.7 | 26.1 | 25.9 | −0.71 | 26.0 | 26.0 | 26.3 | 26.1 | −0.60 |
| Mist and 600 W | 23.5 | 23.5 | 23.6 | 23.7 | 23.9 | 23.7 | −0.74 | 23.6 | 23.8 | 23.8 | 23.7 | −0.74 |

Both sets of sensors show an average temperature drop from misting of 0.6–0.7 K. The use of one 300 W heater increased the average temperature by 1.0 K, use of 600 W heating increased the temperature by 2.5 K. A combination of mist and 300 W heating yields a net increase of 0.3 K, while 600 W heating yields maximum a net increase of 1.8 K.

*3.2. Thermal Sensation Evaluation Experiments*

Each experiment took place over two (Trials 1, 3, Supplement trials 1 and 2) or four (Trial 2) days. The fan used to create VMAS was switched each day, without moving any other equipment or sensors. This should have helped counter bias in the case of measurement error.

Experiment subjects were all volunteers recruited from passers-by at a university campus. Thus, there may be selection bias in that people who expect the mist to be unpleasant chose not to participate. No elimination criteria were used to select subjects. All who volunteered to participate were accepted. Experiment trials 1, 3 and 5 were conducted in summer during open campus events, thus most subjects were high school students visiting the campus, often with their families. Experiment trial 2 and supplement trial 1 were conducted in autumn in a university laboratory, thus most subjects were university students and a few staff. The college has a male/female student ratio of about 30/70, and there was a similar bias in gender of participants in all trials, as detailed in Table 4.

**Table 4.** Summary of experiment trial parameters and subjects.

| Trial | Blinded? | Season | Days | Subjects | Male | Female | Age (Avg $\pm$ SD) | Max Age | Min Age |
|-------|----------|--------|------|----------|------|--------|--------------------|---------|---------|
| 1 | Yes | Summer | 2 | 151 | 18 | 133 | $19 \pm 11$ | 60 | 10 |
| 2 | Yes | Autumn | 4 | 99 | 28 | 71 | $21 \pm 4$ | 47 | 18 |
| 3 | Yes | Summer | 2 | 53* | 13 | 40 | $22 \pm 13$ | 67 | 13 |
| Supp. 1 | No | Autumn | 2 | 64 | 13 | 51 | $24 \pm 9$ | 68 | 19 |
| Supp. 2 | No | Summer | 1 | 63 | 17 | 46 | $22 \pm 5$ | 53 | 18 |

* 142 subjects participated, but temperature data was lost for all but 53, see below.

Test subjects were exposed to the air streams from the two identical misting fans, clearly labelled "Fan A" and "Fan B" and placed side-by-side on a table such that the air stream height was at 1.3 m. One fan had the mist function active, and one did not. The subjects were asked to judge the thermal sensation with their hands. Subjects were told the experiment was to obtain data on the cooling effect of fans with or without MEC. They were not told about the heater nor the possibility that the misted air was warmed.

Each subject was given a paper ballot with three choices, "Fan A", "Fan B" or "Same", as well as items on age, sex and time at which the trial was conducted. Subjects were allowed to take as much time as desired. All made a judgment within 30 s. Temperature data were recorded at 10-s intervals. Two sensors are mounted above traffic cones at 50 cm distance along the centerline of the blown air to record the temperatures of the VMAS and FAS. The difference between these two sensors is the $\Delta T$. This setup is shown in Figure 3, where the "X" marks show the position of the temperature sensors. The temperature of the VMAS and FAS during each evaluation was found by matching the recorded evaluation time on the subjects' survey form to the recorded temperature data, time averaged over that 1 min.

The first trial was a proof-of-concept experiment to determine if a VMAS would be judged as cooler when heated to about the same temperature as the FAS. This seemed to be true for most respondents (as detailed below in the results). Thus, a follow-up trial was conducted over a wider range of temperatures, such that the VMAS was up to about 2 K warmer than the FAS. A third trial was conducted outdoors during the following summer, focusing again on equal or higher temperature in the VMAS.

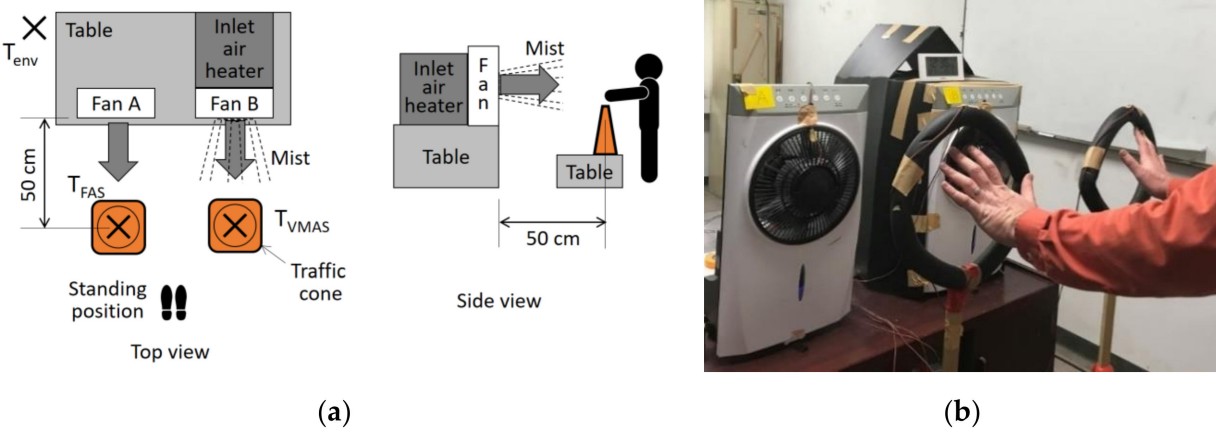

(**a**)  (**b**)

**Figure 3.** (**a**) Setup of the misting fan experiment. Three "X" marks indicate sensor positions. (**b**) photo of indoor setup and hand placement guide hoops (added for the second and all further trials).

Outdoor trials were conducted in a well-shaded area at the covered entrance area of the college in Osaka, Japan, at what is typically the hottest period of the year, early August. Indoor trials were conducted in a non-conditioned room with open doorways to the hall during the autumn when temperatures were typically around 20 °C.

### 3.3. Particulars of the Three Blinded Trials

The first experiment trial was conducted outdoors. Subjects were recruited from visitors to a university open campus event on 6–7 August 2016, between the hours of 10:30 to 15:00 each day. The weather on each day was sunny and among the hottest days of the year, the nearest meteorological stations at Sakai and Osaka (about 4 km and 7 km distant) recorded a high temperature of 35.5 °C on 7 August with humidity ranging over 44–47%RH and 35.4 °C at 47–51%RH on 8 August. The recent thermal history of subjects is assumed as mostly having been outdoors and walking while touring the campus, with a smaller fraction having been in indoor explanatory sessions where air conditioning was set to 28 °C in accordance with Japan's "Cool Biz" energy conservation policy.

In this trial, on the first day, some subjects evaluated the streams with their hands very close to the fan, within the VMAS mist cloud. This may have affected the thermal condition due to contact with mist droplets. Subjects were instructed to not directly touch the mist on the second day.

The second trial was conducted the same as the first trial. The heating system was adjusted to create a wider range of temperature difference between the VMAS and FAS, to exceed the possible range of measurement error. The VMAS was warmed up to about 2 K higher than the FAS. To increase uniformity of the evaluation and prevent the possible influence of wetting by touching the mist cloud, 40 cm diameter guide hoops (see Figure 2, right) were mounted 50 cm from the mist fans along the centerlines of the air streams. Subjects were instructed to place their hand anywhere within the guide hoop, maintaining a 50 cm distance from the fans. The trial was conducted indoors over 4 days in October and early November. The misting fan and non-misting fan were switched each day. The room temperature ranged from 18–22 °C with humidity ranging around 40–60% on those days, similar to the outdoor temperature. This is a much lower temperature than during the typical use of MEC systems. The recent thermal history is assumed as students leaving a classroom (non-conditioned), those on the way to class were too rushed to participate.

The third trial was conducted in the summer of the following year using the guide hoops in hot summer conditions. The heating of the misted air stream was performed with two 300 W electric-resistance heaters. In total, 142 subjects participated. However, the temperature log was not saved for the more than half of the subjects due to operator error. Thus, temperature data are only available for 53 of the test subjects. Only these data were evaluated for this trial. These data are only from one day, thus the daily switching of fans

as a counter to experimental error is not available. The weather on both days was sunny and among the hottest days of the year. The Sakai meteorological station recorded a high temperature of 36.4 °C on August 5 with humidity ranging at 52–59% RH and 35.9 °C at 42–53% RH on August 6.

### 3.4. Procedure of the Supplemental Unblinded Experiment Trials

The primary experiment uses deception as a form of blinding (by hiding the heaters) to the fact that the VMAS may be warmer than the FAS. If subjects are told about the heaters, would trend in responses change?

The procedure was the same as the second and third trials, with the exception that the heaters were not hidden. Subjects were all notified of the heating system behind the misting fan. They were told the heater was a means to help control the temperature range of the VMAS. The subjects were told that this was a test of comparing the response of human skin to the temperature sensors, which may have a tendency to yield error due to possible wetting. Although that concept is true, the sensors were not wetted in these experiments.

The first supplemental trial was performed outdoors on 4 and 5, August in the same location as Trials 1 and 3. The high temperatures at the nearest meteorological station on the two days were 35.7 °C and 36.1 °C with humidity ranging from 40–45% on both days. The second supplemental trial was performed indoors in similar conditions to Trial 2 on 31 October with temperatures ranging from 17–20 °C and humidity of 41–48%.

## 4. Results

The null hypotheses are evaluated over the range of VMAS-FAS temperature difference in the experiments (in Trials 2 and 3 the VMAS is sometimes up to about 1.6 K warmer than the FAS), to account for the range of measurement error and test the possibility that VMAS is chosen as cooler even when it is warmer than the FAS.

The first trial was over a relatively narrow range of temperature difference between the VMAS and FAS, $\Delta T$. Positive values indicate that the VMAS was warmer. Figures 2a, 3a and 4a show the distribution of temperature difference $\Delta T$ in box plots for each of the votes of "mist", "same" and "fan only" as to which air stream felt cooler. The values of $\Delta T$ are time-averaged over the 1 min during which the subject evaluated the air streams. The number of responses in each bin is noted above each bar in the graph as "$n$".

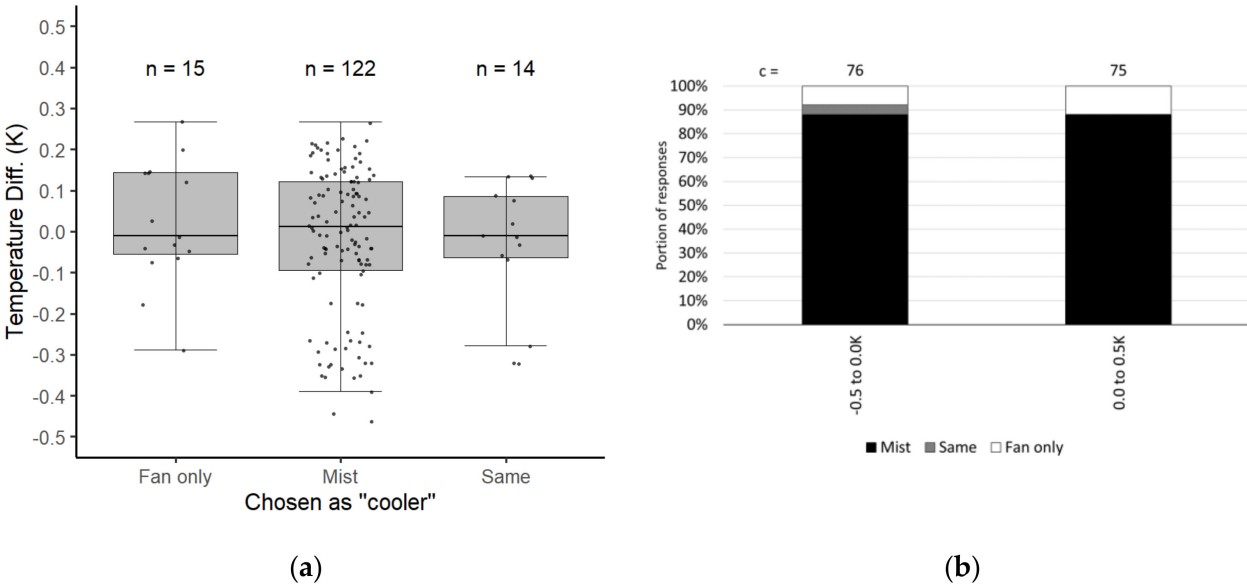

(**a**)                                                                                (**b**)

**Figure 4.** Distribution of temperature differences (**a**) and responses (**b**) from the first experiment trial.

In the first trial (Figure 4a), $\Delta T$ ranged from $-0.47$ K (the mist was actually cooler) to $+0.27$ K (the mist was 0.27 K warmer than the "fan only" case). The average $\Delta T$ is nearly the same for the 3 cohorts. About 80% of the subjects chose the VMAS in this range.

The second trial (Figure 5a) extended the $\Delta T$ up to $+1.61$ K, with a minimum of $-0.20$ K. There is an even split of subjects choosing "fan only" and "mist" as coolest. There is a clear trend that "fan only" is chosen as cooler more often when $\Delta T$ is high (the FAS actually is cooler than the VMAS).

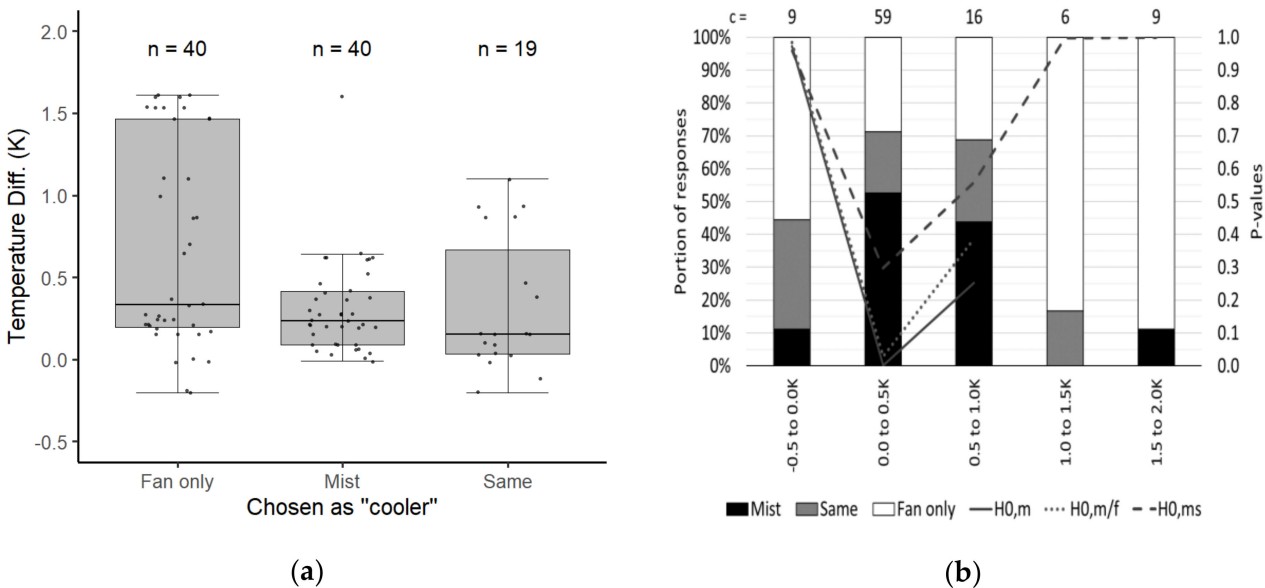

(**a**)                                          (**b**)

**Figure 5.** Distribution of temperature differences (**a**) and responses (**b**) from the second experiment trial.

The third trial (Figure 6a) focused around $+0.15$ K $< \Delta T <$ $+1.07$ K. The votes for "mist" are most common. Again, there is a slight trend for votes of "fan" at higher temperature difference.

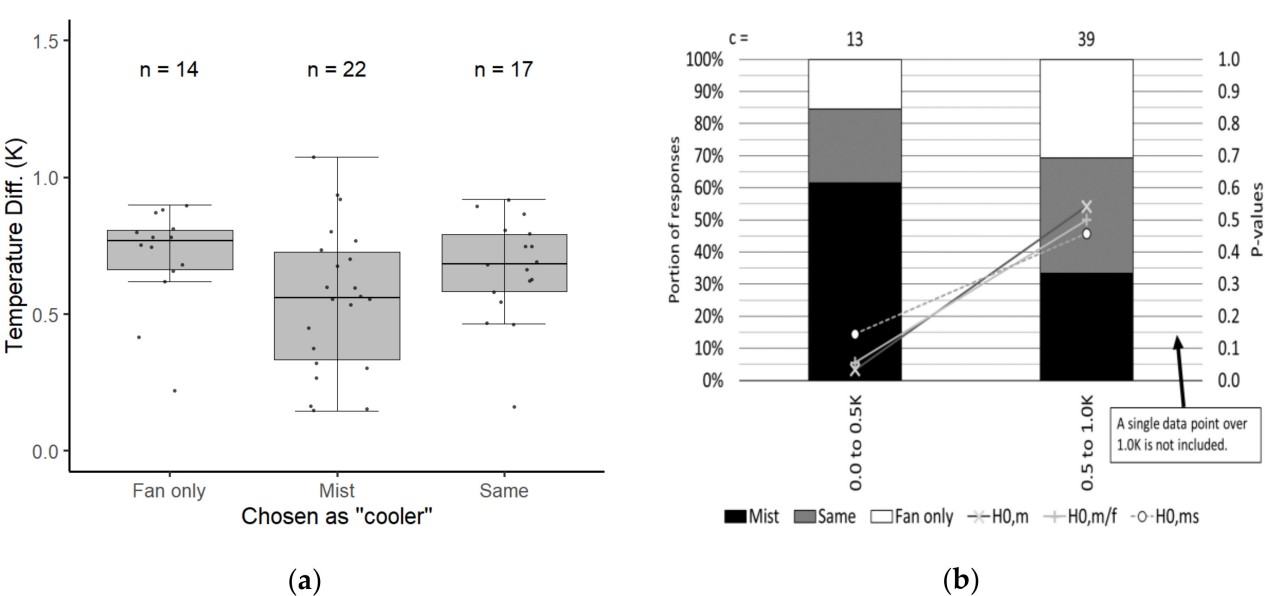

(**a**)                                          (**b**)

**Figure 6.** Distribution of temperature differences (**a**) and responses (**b**) from the third experiment trial.

The fractions of responses for VMAS ("Mist"), FAS ("Fan only") or "Same" for bins of 0.5 K in temperature difference are shown as stacked bar graphs in Figures 2b, 3b and 4b. The three null hypotheses are tested in each bin as compared to a cumulative binomial distri-

bution. The three null hypothesis are labelled as $H_{0,s}$ (< 50% choose "Same"), $H_{0,m/f}$ (< 50% of those not choosing "Same" choose mist), and $H_{0,ms}$ (< 67% choosing mist or "Same"). The total number of responses in each bin is labelled as "c" at the top of each graph.

In the first trial (Figure 4b), the three null hypotheses yielded $p < 0.001$ for all three. Thus, all three nulls are rejected where the temperature range is fairly narrow from $-0.5$ K through 0.5 K temperature difference. The VMAS and FAS are about the same temperature, but subjects overwhelmingly chose the VMAS as cooler.

In the second trial (Figure 5b), the first and second null hypotheses have $p < 0.05$ for the 0.0–0.5 K bin. $p$-values are higher in all other bins, but all other bins have far lower numbers of responses. Thus, the first and second null are rejected for the 0.0–0.5 K range, not rejected for the 0.5–1.0 K range, and no conclusion can be made from the other bins, though there is a trend to not choose the VMAS. The third null hypothesis cannot be rejected at any temperature range.

In the third trial (Figure 6b), the first and second null hypotheses are at $p \leq 0.05$, while the third null is at $p = 0.14$ in the 0.0–0.5 K bin. In the 0.5–1.0 K bin only the first null can be rejected. Note that 1 data point fell above 1.0 K. It is not included in the graph.

When using a bin width of 0.5 K, it seems that for the temperature difference span of 0.0–0.5 K, where the VMAS is the same temperature or slightly warmer than the FAS, there is a bias to choose VMAS as cooler. As the temperature difference becomes larger, the trend falls, though some people will still choose the VMAS as cooler.

To check that these results are not due to a coincidental alignment of the bin width and limits to the data (i.e., Do the results change if the 0.0–0.5 K bin was instead 0.1–0.6 K?) and explore the trend in responses over the range of temperatures in the experiments as well as account for the error in $\Delta T$ (about $+/-0.4$ K), the responses and batched into cohorts of 0.5 K spans (bin width). We then shift the 0.5 K cohort span in steps of 0.1 K to include the whole range of temperature differences in each trial. Thus, some members of each bracket are also included in neighboring brackets. The count of responses in each cohort is here termed as $c$ to avoid confusion with $n$, the number of subjects in each trial. The portion of responses choosing "Mist", "Same" and "Fan only" in each cohort are shown in stacked bar graphs for each trial as Figures 5–7.

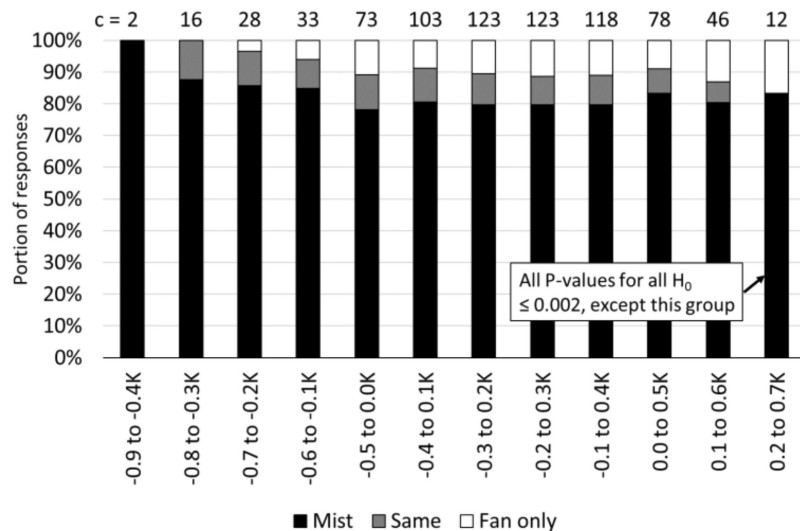

**Figure 7.** Proportion of responses on which air stream felt cooler in a rolling evaluation of 0.5 K bin width steps. Responses from the first experiment trial.

In the first trial (Figure 7), there is a clear majority to choose mist as cooler in all cohorts. However, most of these cohorts include temperature difference where the mist is actually cooler. The resulting $p$-values are under 0.001 for all three null hypotheses in each cohort. Thus, the three null hypotheses are rejected. People would likely choose that this

mist is cooler over this temperature range. A majority of those who did not choose "same" would choose this mist. Finally, a majority of people would choose mist or "same" for this VMAS to FAS comparison in all temperatures. Noting that some people put their hands in the mist fan the first day, this analysis was performed again using only the data from the second day. The results were nearly the same, with a slight reduction in the choices of "same", but all $p$-values again under 0.001.

In the second trial (Figure 8) most of the $\Delta T$ brackets have the VMAS as warmer than the FAS. There is a trend to choose VMAS at about 40–60% in the brackets up to the 0.6–1.1 K bracket (where the VMAS was actually a bit warmer, beyond calibrated measurement error, but just within the rated measurement error). The $p$-values for the first null hypothesis are under 0.05 for the brackets from $-0.2$–0.3 K through 0.3–0.8 K. Examining the second null (those who did not choose "same") the $p$-value is under 0.05 in 3 cohorts in the same range as the first null. $p$-values are fairly low in many other cohorts indicating a trend, but not meeting a 0.05 level. The third null (choice of mist or same <67%) cannot be rejected at any temperature range.

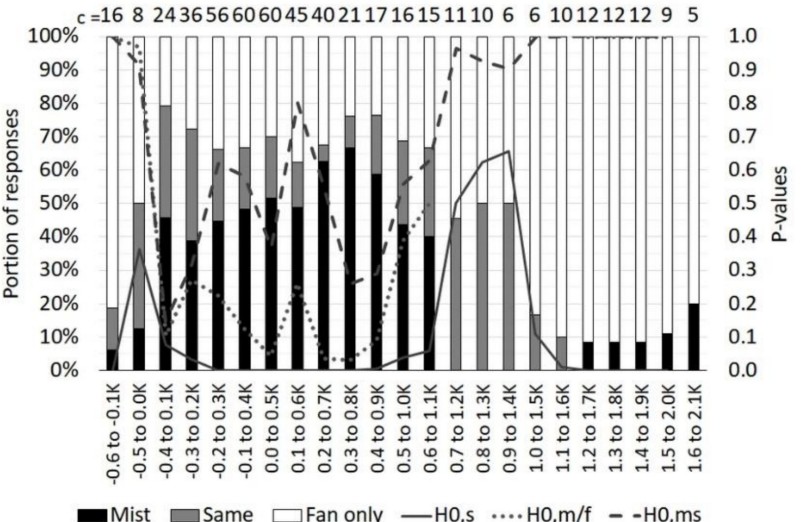

**Figure 8.** Proportion of responses on which air stream felt cooler in a rolling evaluation of 0.5 K bin width steps. Responses from the second experiment trial.

Overall, the second trial results indicate some bias toward choosing VMAS as cooler or the same as FAS, even when it was measurably warmer to a small extent. The range of temperatures is wider and warmer than the first trial. There is an unexpected result in the coolest two cohorts at the left side of the figure in which few chose mist, where the mist actually was slightly cooler. However, the cohort numbers are relatively small (c = 16, c = 8).

In the third trial (Figure 9), there is a trend to choose mist as cooler starting around 70% which drops steadily as the $\Delta T$ increases (not including the final 2 cohorts at right, where the cohort counts are only 4 and 1). The $p$-values for the first null (> 50% for same) are under 0.05 for all cohorts up to the 0.1–0.6 K level, with $p = 0.07$ at the next cohort. The second null hypothesis (<50% mist for those not choosing "Same") has $p < 0.05$ for all cohorts up to the 0.2–0.7 K cohort. The third null (<67% choosing mist or "Same") can be rejected for most brackets up to the 0.6–1.1 K bracket.

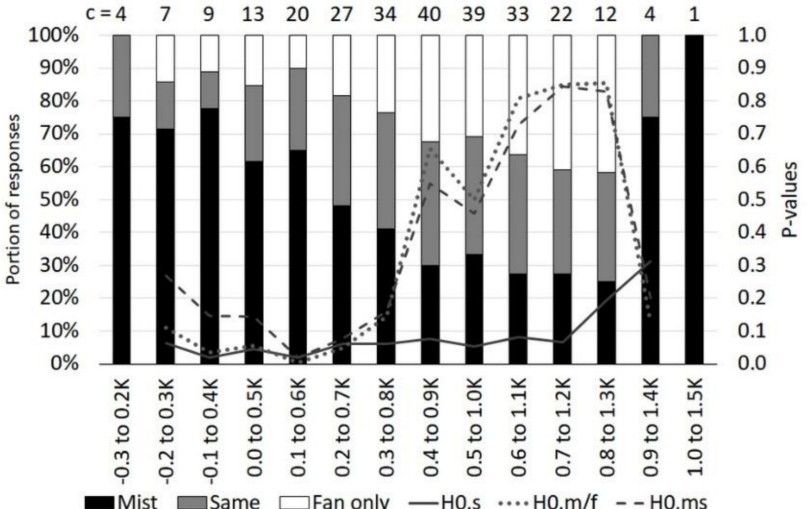

**Figure 9.** Proportion of responses on which air stream felt cooler in a rolling evaluation of 0.5 K bin width steps. Responses from the third experiment trial.

As with the second trial, there is a trend to choose mist as cooler. The first and second null hypotheses can be rejected up to 0.1–0.6 K or 0.2–0.7 K, which is about the range of calibrated measurement error, and within the range of rated measurement error. The third null could be rejected for only one $\Delta T$ cohort at 0.1–0.6 K.

*Results of the Supplemental Unblinded Experiment Trials*

The number of test subjects in the two supplemental trials was considerably smaller than in the primary experiments. Thus, the statistical check of the three null hypotheses has less weight than in the primary experiments. Further dividing the brackets into rolling 0.1 K steps (as in Figures 6–8) would yield even smaller cohort numbers, and thus is not done here.

In the first supplemental trial, Trial 4, (Figure 10), the first null is rejected only when the VMAS was actually cooler. No other nulls were rejected in any other temperature range. In the $\Delta T = 0.0 \sim 0.5$ K range, there is a clear trend to choose "same" and an even split between VMAS and FAS. A trend to choose FAS as cooler increases as the temperature difference $\Delta T$ increases, but the number of responses in each bracket is quite small. Overall, the votes seem to better match reality than in the primary trials.

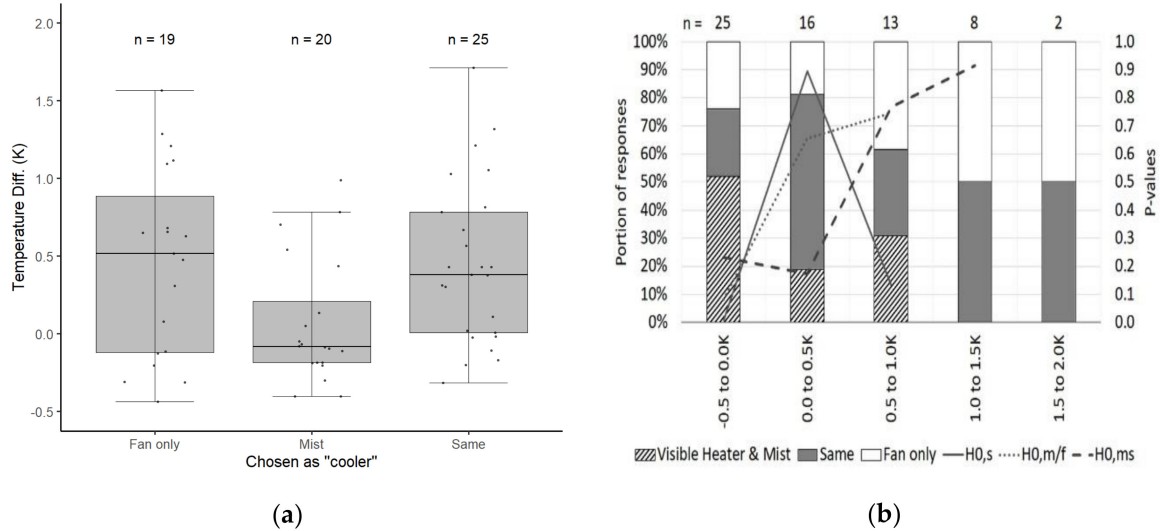

(**a**)                                                                                               (**b**)

**Figure 10.** Distribution of temperature differences (**a**) and responses (**b**) from the first supplemental trial, Trial 4.

In the second supplemental trial, Trial 5 (Figure 11), the first null could not be rejected in any temperature difference range, though about 34–40% of responses were for "same" in the first two brackets. There was a trend to choose VMAS at lower $\Delta T$. However, the second and third nulls might be rejected in the $\Delta T = 0.5$–1.0 K bracket and nearly so in the 0.0–0.5 K bracket. That is, there was a strong trend to not choose the FAS as cooler among those who did not choose "same", as well as a trend to not choose the FAS as cooler at any bracket even though it actually was. However, the number of subjects in this trial is quite low. The nulls could not be rejected in the highest bracket of 1.5–2.0 K, though there were only 12 responses.

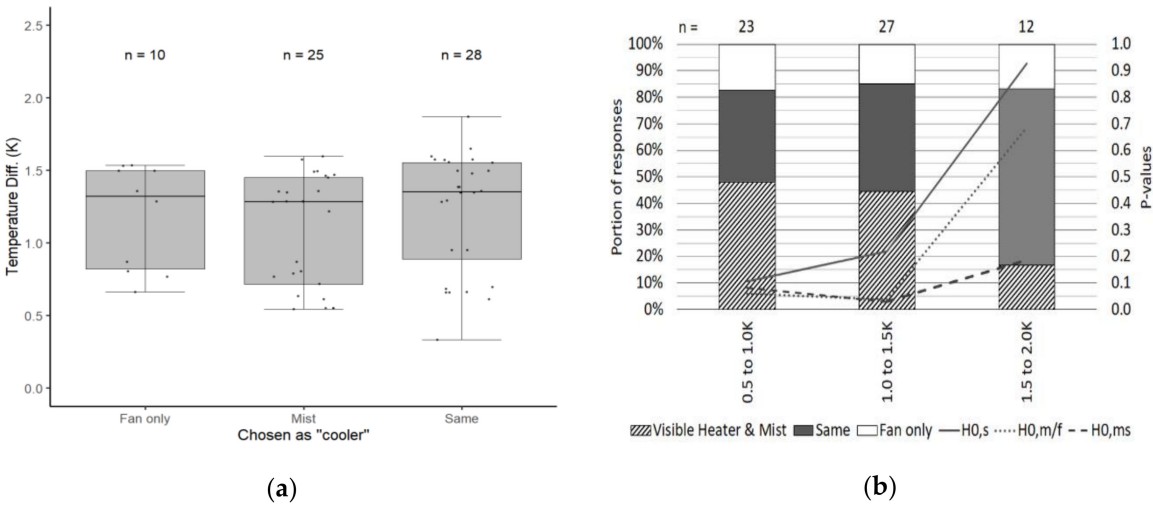

(**a**)　　　　　　　　　　　　　　　　　　　　　　　　(**b**)

**Figure 11.** Distribution of temperature differences (**a**) and responses (**b**) from the second supplemental trial, Trial 5.

A summary of the statistical results for the 0.5 bin widths is shown as Table 5. Overall, there is a trend to bias for VMAS as cooler in the blinded trials for the 0.0~+0.5 K range, where the VMAS is slightly warmer. In the unblended trials there is no trend during the autumn trial, but there is still a weak trend in the summer trial.

**Table 5.** Summary of experiment trial statistical significance.

| Trial | Temperature Difference, $\Delta T$ | | | | | | | | | | | |
|---|---|---|---|---|---|---|---|---|---|---|---|---|
| | −0.5~0.0 K | | | 0.0~+0.5 K | | | +0.5~+1.0 K | | | +1.0~+1.5 K | | |
| | $H_{0,s}$ | $H_{0,m/f}$ | $H_{0,ms}$ | $H_{0,s}$ | $H_{0,m/f}$ | $H_{0,ms}$ | $H_{0,s}$ | $H_{0,m/f}$ | $H_{0,s}$ | $H_{0,s}$ | $H_{0,m/f}$ | $H_{0,ms}$ |
| 1 | *** | *** | *** | *** | *** | *** | | Not tested | | | Not tested | |
| 2 | x | x | x | *** | * | x | * | x | x | x | x | x |
| 3 | | Not tested | | * | * | x | * | x | x | x | x | x |
| Supp. 1 | ** | 0.08 | x | x | x | x | x | x | x | x | x | x |
| Supp. 2 | | Not tested | | | Not eval. (*n* = 1) | | x | 0.06 | 0.08 | x | * | * |

Note: * $p \leq 0.05$, ** $p < 0.01$, *** $p < 0.001$, x $p > 0.1$, $p$ values between 0.05 and 0.10 are shown.

The experiment data and corresponding figures and statistical analysis are available as noted in "Supplementary Materials" at the end of this paper.

## 5. Discussion

In the primary trials (1–3), the test subjects showed a tendency to vote a VMAS as cooler than a FAS even when the VMAS was about the same temperature as the FAS. The trend is quite close to the possible range of error in $\Delta T$, $+/-1.0$ K as rated, $+/-0.4$ K as calibrated on site. However, the cooling effect of this thin mist spray will reduce the average air stream temperature by only 0.4 K, or locally by 0.7 K along the centerline. The minimum temperature difference in all trials was $-0.47$ K, which would correspond to a

period in which the heating system was off and the VMAS cooling took full effect. It is unlikely there is an error such that the temperature difference is actually lower than the measured values presented in the results.

The trend was present both in hot summer and moderate autumn conditions. However, the second trial, conducted in autumn, showed a lower trend to choose VMAS. Seasonal acclimation might have been an influence. A lack of desire for cooling may have been an influence. There is no thermal stress during the mild conditions of the autumn. Environment air temperature was lower than summer. However, the VMAS and FAS are of about the same temperature, be it summer or autumn.

Switching the fans each day and calibrating the sensors with each other reduce the chance that there was a systematic temperature error in the experiments.

### 5.1. Discussion of the Supplemental Unblinded Experiment Trials

The results of the first supplemental trial, performed in summer, presented a more realistic evaluation of the VMAS and FAS with their actual temperature. When the VMAS was cooler, it was chosen as cooler. When the temperature was the same or slightly warmer, "same" was chosen. At higher temperatures, there was a trend to choose the FAS.

The sight of a glowing red heater element during the hot summer day might yield an expectation of warmth to counteract the expectation of coolness from the visible mist. The knowledge that the VMAS might be the same or higher temperature than the FAS is a likely influence to reduce a bias to choose the VMAS as cooler. The subjects of the preliminary trials had no reason to expect the VMAS might be the same or warmer temperature than the FAS.

The more realistic evaluation did not continue in the autumn experiment, where the choice of FAS was quite low in all temperature brackets. In the relatively cool autumn, expectation, or perhaps preference, may be different than in summer.

### 5.2. Discussion of Possible Implications of the Results

The concept of thermal alliesthesia [42] is that a change in thermal environment from an unpleasant condition toward a neutral condition (from hot to cooler, from cold to warmer) is evaluated as pleasant. The convection cooling of a fan in summer presents a transition from hot to neutral. Adding mist increases the effect. People will likely have an expectation that fans and mist are pleasant. The convection cooling of a fan during the neutral or slightly cool temperature of autumn is a driver away from neutral toward the cold, and would yield an evaluation of unpleasant.

Here, the survey question is not asking for a comfort evaluation, but simply which fan is cooler. The alliesthesia concept may not directly apply to such an evaluation, especially considering that the VMAS and FAS have identical thermal characteristics except the slight temperature difference. However, alliesthesia may be a factor affecting the perception or expectation of fans and mist in different seasons (summer or autumn). If this experiment were performed in the cold of winter, it might yield quite different results.

The supplemental trials revealed the use of the heater. The heater might produce the reverse expectations of the fan and mist, a strong expectation of unpleasantness in hot summer conditions, and a pleasant expectation in the slightly cool autumn. These conflicting expectations may have influenced the change in evaluations between the summer and autumn trials, resulting in a more balanced or accurate evaluation in summer, while in autumn there was a trend to choose VMAS as "same" or cooler.

Although the research goal is to determine if there is a bias to perceive mist as cooler, this perception may not only be due to expectation. Subjects may have been motivated to be "good subjects". We took care to avoid overt cues and did not tell subjects the true experiment hypothesis. However, the test question itself, "Which feels cooler?" may push the participant to choose one of the two air streams, even though they were explicitly given the option to vote for both streams as feeling the same. Some participants may have guessed that the goal of a typical mist comfort experiment was to find that mist

feels cooler. If they are "good subjects" they might give the "desired" result, voting for mist as feeling cooler, rather than voting for their honest perception. Further, the physical appearance of the experimenters during the trials on hot summer days may have been a cue. Experimenters cannot help but show clear signs of thermal stress such as sweating and fatigue, which may have provoked sympathy for the experimenters. Subjects themselves also show signs of thermal stress, which may influence nearby subjects. When indoor tests were performed in autumn, the experimenters were not under thermal stress, which might reduce such a sympathetic response.

The indoor experiment participants in autumn (Trials 2 and 5) were almost all students in the college in which research on mist cooling has been conducted for several years. Campus rumors may have influenced the trend to vote as "good subjects" in favor of mist as cooler (or the opposite, if they happen to not like the experimenters). On the other hand, the summer experiments were nearly all visitors who had not been to the university before. They likely had not heard of this type of experiment. Thus, rumor was likely not a factor in the summer results (Trials 1–4).

There may be other influences at work, such as a possible difference in the smell of misted air. However, such an evaluation is beyond the scope of this experiment.

Typical outdoor mist cooling installations can yield temperature drops of 1–5 K or more, which is far in excess of the temperature difference range of these experiments. The improved comfort votes in tests of mist are likely not explained as being entirely the result of the possible expectation effect found here, which may include "good subject" effects. As the evaluations were performed only by feeling the VMAS and FAS with the hands, the sudden temperature change of a small region of skin causing "spatial alliesthesia" [42] is also a likely factor, to be explored in future experiments over longer time periods.

Further experiments are planned, especially in the higher range of temperature difference (the misted air stream warmed 0.5–2 K above the non-misted stream). The temperature control must be improved, and care must be taken to get more votes at the extremes of the temperature range explored here. Subjects should be asked about their knowledge and experiences with mist cooling to check for possible influence. Blinded experiments should be performed, in which the subjects cannot see the misted air stream in evaluating it against a non-misted air stream.

*5.3. Discussion of Limitations of this Experiment*

As this is a fairly new experiment for this study of MEC, we performed many trials with slight adjustments to the methods as the work progressed. This non-uniformity yielded a scattering of the temperature conditions, with more evaluations around $0 < \Delta T < +0.5$ K with reduced numbers at the tails, where the VMAS was actually cooler or warmer. This yields less statistical confidence in the results at the tails of the temperature distribution. Furthermore, conducting trials in both summer and autumn makes it difficult to compare the results of all trials. However, there were interesting differences between the two seasons which hint at a seasonal effect.

As the experiment depended on presenting VMAS and FAS of about the same temperature with no wetting, the easiest means to accomplish this was with a very thin mist that itself only yields about 0.5 K of cooling. However, many MEC deployments create temperature drops of around $2-3$ K or more. A future experiment that compares a more typical MEC system with dry air of the same temperature, or some other form of blinding to evaluate the possible perception bias is desirable.

There is a selection bias in the test subjects. Those who might expect the MEC to be unpleasant may have chosen not to participate. However, this would likely be the same as in application of MEC for outdoor cooling. People who expect displeasure would more likely avoid MEC if possible. (This issue should be considered in any MEC deployment in public use. There should be a route or area for those who want to avoid MEC.) There was no rejection of any subjects who chose to participate. Furthermore, the great majority of subjects were female.

The judgment of MEC was performed solely by bare hands. A future experiment should test evaluations of MEC for the whole body. However, that would make it difficult to evaluate both misted air and non-misted air at the same time. A step change and the timing of it would become a factor.

## 6. Conclusions

The results indicate that there may be some bias to perceive a VMAS to be cooler than it actually is. Three null hypotheses were tested;

1. $H_{0,s}$: When comparing a VMAS to a FAS of similar temperature and velocity, given three choices (VMAS, FAS, or "Same") more than 50% of subjects will (correctly) evaluate them as the same

2. $H_{0,m/f}$: When comparing a VMAS to a FAS of similar temperature and velocity, among those who do not choose "Same", no more than 50% will choose the VMAS as cooler.

3. $H_{0,ms}$: When comparing a VMAS to a FAS of similar temperature and velocity, no more than 67% will choose the VMAS as cooler or "Same".

The first null could be rejected for most temperature differences up to the cohorts where the VMAS ranged up to 0.6 K (Trial 2) or 0.8 K (Trial 3) warmer than the FAS. This exceeds the calibration measurement error. Thus, we believe that people will not correctly evaluate a VMAS and FAS of the same temperature as being thermally the same, and this may even tend to be true when the VMAS is slightly warmer.

The second null could be rejected for some temperature differences up to the cohort where the VMAS was 0.3–0.8 K warmer than the FAS. This does not entirely exceed the calibration measurement error. It is likely that when temperatures are the same, among those who do not choose "same", people will tend to choose mist.

The third null could only be rejected for most temperature differences up to the cohort where the VMAS was 0.5–1.0 K warmer than the FAS, but only for the trials performed in summer. It could not be rejected for any temperature range in the autumn trial.

Most of these ranges exceed the on-site calibrated measurement error, and nearly meet the worst case of manufacturer rated measurement error. The number of responses in each 0.5 K bin width cohort was quite small for many of the temperature differences at the cool and warm extremes. Further experiments with more subjects and stricter temperature control are required.

In short:

1. There is a bias to perceive VMAS as cooler than a FAS of the same temperature, or slightly higher.

2. Results may depend on the season. The trend to this bias for VMAS as cooler is stronger in summer than in autumn. When a heating element was revealed behind the VMAS, in autumn there was an unexpected bias against the FAS as cooler.

This likely mist over-evaluation effect could account for some of the tendency for thermal sensation votes in mist to overshoot the predictions of thermal comfort models such as PMV. In creation of databases of human evaluations of mist cooling to create predictive physical models, it may be necessary to blind the experiments to get votes without a possible mist cooling expectation effect.

This research does not determine the cause of this effect—be it expectation, the "good subject" effect, or something else. However, if it is due to expectation, deployment of mist cooling systems for thermal comfort could exploit this by ensuring that the mist spray is visible to the persons being cooled.

**Supplementary Materials:** The followings are available online at https://www.mdpi.com/2571-879 7/3/1/11/s1, Figures 2 and 4, Figures 5–11 and Tables 3–5 as Excel files, and the R program used to create the boxplots in Figures 4–6, 10 and 11.

**Author Contributions:** Conceptualization, investigation, data curation, writing–review and editing, C.F. and J.Y.; Methodology, formal analysis, supervision, visualization, writing–original draft, C.F.; All authors have read and agreed to the published version of the manuscript.

**Funding:** This research received no external funding.

**Institutional Review Board Statement:** Ethical review and approval were waived for this study, due to the study being a voluntary anonymous preference survey in which no personal identifying data was taken.

**Informed Consent Statement:** Subject consent was verbal and all subjects were voluntary and free to refuse participation. There was no measurement or personal data collection, nor any influence on the condition of the interviewed subjects. The study was a simple anonymous preference survey, similar to a commercial product survey.

**Conflicts of Interest:** The authors declare no conflict of interest.

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
