# Peer review of "Possible Perception Bias in the Thermal Evaluation of Evaporation Cooling with a Misting Fan"

_cleantechnol, doi:10.3390/cleantechnol3010011_

Round 1

Reviewer 1 Report

The paper idea is outstanding and relevant. The authors challenge the expectations of mist evaporative cooling as a technology. The research hypothesis is clear and the results are clear. However, the paper suffers from a weak positioning regarding the state of the art and the methodology section is poorly written and does not allow for reproducing the experiments. I invite the authors to address the following comments carefully in order to make this paper publishable:

  1. Introduction: Please write a classical introduction with a problem statement, research objectives, and research questions. The authors must also mention the added value of the work and its significance and expected audience that might benefit from the study.
  2. A whole section called stated of the are or literature review should be added. The latest and most recent paper must be cited. The similar studies not only the thematic once but also the similar methodological studies must be cited. The overestimation of the mist evaporation cooling technology in other studies must be very well documented and explained.
  3. The methodology section is very poorly written and is very confusing. There are parts of the study the survey design, sampling process, and recruitments process. Then the experiments and their settings and the parametric variation. The metrology failed to explain the first part and described the second part in too much detail. It is not clear who was recruited, when, and how. The authors are so obsessed with describing the experiment while neglecting the description of the observational methodology of the survey sample.
  4. The experiment goals and questions must be moved to the introduction.
  5. It is not clear how the researcher avoid bias and how did they validate their questioning approach. What statistical test was conducted.
  6. The whole experimental procedure and protocol should be re-written under one subheading.
  7. All graphs in the results section should be modified to exclude the frame.
  8. Results must be reported separately.
  9. The Discussion must be reported separately in a discussion section.

Reviewer 2 Report

The title of the article needs to be changed into something more appropriate.

Introduction:

  • Need references in the intro. Consider writing an intro that provides a better justification for the problem examined. Currently, the introduction is short.

Literature review:

  • A lot of the literature reviewed does not consider recent articles. Is it because the topic is vaguely investigated? I suggest incorporating post-2015 studies

Experimental Design:

  • It would be better if a summarising figure of the method adopted is presented to the readers
  • The null hypothesis examined is not justified. What are the reasons behind posting such questions.
  • Line 185: “Further, this study does not seek to quantify the possible over-evaluation of mist cooling”. Not sure what you mean by that. Please elaborate further

Results:

  • A table reporting statistical observations and tests of significance is essential

Conclusion:

  • There are several limitations that need to be reported. Please add at least 3 at the end

Round 2

Reviewer 1 Report

Accepted.